# Development, Implementation, and Evaluation of a ‘Virtual Patient’ with Chronic Low Back Pain: An Education Resource for Physiotherapy Students [note 1]

**DOI:** 10.3390/healthcare13070750

**Published:** 2025-03-27

**Authors:** Kate Thompson, Steven Bathe, Kate Grafton, Niki Jones, David Spark, Louise Trewern, Thomas van Hille, Mark I. Johnson

**Affiliations:** 1Centre for Pain Research, School of Health, Leeds Beckett University, Leeds LS1 3HE, UKm.johnson@leedsbeckett.ac.uk (M.I.J.); 2Harrogate and District NHS Foundation Trust, Harrogate HG2 7SX, UK; 3Independent Researchers, Leeds LS1 2HE, UK; 4Centre for Learning and Teaching, Leeds Beckett University, Leeds LS1 3HE, UK; 5Leeds Teaching Hospitals NHS Trust, Leeds LS9 7TF, UK

**Keywords:** pain, biopsychosocial, person-centred, multidisciplinary, physiotherapy, simulation, education

## Abstract

**Background:** The management of chronic pain is inherently multidisciplinary, requiring collaboration across health and care professions because pain is multidimensional, involving psychological, social, biomedical, cultural, and environmental factors. However, pain education has often focused more on biomedical aspects, limiting the capacity of professionals to deliver integrated, person-centred care. Shifting pain education away from biomedically driven curricula may better prepare graduates for meaningful consultations and biopsychosocial care. **Objective:** This manuscript reports the development and pilot evaluation of a virtual patient simulation designed to help physiotherapy students develop person-centred pain assessment skills. Methods: We developed and piloted a virtual patient with complex pain scenarios for physiotherapy students. To evaluate the simulation, students completed a self-reported questionnaire assessing their ability, self-confidence in person-centred assessment skills, and their attitudes and beliefs regarding the simulation. **Results:** Frequency and confidence in person-centred inquiry ranged from 100% to 16.3%, depending on the complexity of information. Inductive thematic analysis revealed four themes: (1) Environmental factors & preferences—students’ preference for the learning environment; (2) Learning experience—including engagement, feedback, discussions, and a ‘safe’ space for building confidence; (3) Professional development—insights into person-centred inquiry, personal biases, and emotional challenges; (4) Limitations—including the desire for more complexity, and technical challenges noted. **Conclusions:** The development of this virtual patient simulation enabled healthcare students to engage with a multidimensional perspective on pain, fostering skills essential for biopsychosocial pain assessment and patient-centred care. Although designed and piloted with physiotherapy students, this model holds potential for broader application across healthcare disciplines.

## 1. Introduction

It is estimated that over 20% of adults live with chronic pain, creating a significant personal burden for individuals along with an economic impact on health and social care systems globally [1,2,3]. One of the key roles of health and care professionals is to support people who are experiencing chronic pain, of which physiotherapists play a significant part. Physiotherapists need a wide range of skills and an excellent understanding of the different factors that contribute to a person’s pain experience and how it impacts their well-being and quality of life. The development of knowledge and skills begins in pre-registration training [4,5].

The International Association for the Study of Pain defines pain as “An unpleasant sensory and emotional experience associated with, or resembling that associated with actual or potential tissue damage.” [6]. This simplistic definition hides the complex and often paradoxical nature of pain as a context-driven experience emerging from neural processing influenced by an interplay between psychological, social, biomedical, cultural, and environmental factors [7]. It is for this reason that a person’s experience of pain is ‘unique’ and ‘personal’ and response to the treatment intervention variable, and prevention, treatment, and management a challenge. It is important that physiotherapists, along with other healthcare students, learn to conceptualise pain as a multidimensional personal experience, so that they are adequately prepared to support individuals in clinical practice.

Pain education in pre-registration training should be designed with input from core competencies [8], curricula, and practical guides [9,10]. However, the uptake of these resources in physiotherapy, and other health professional training, appears to be poor [4]. Often, pain education is fragmented and dominated by a biomedical rather than psychosocial focus [4,11]. Physiotherapists need bio-psycho-socio-cultural-environmental pain education to be able to hold meaningful consultations with people experiencing pain, to be able work effectively across multidisciplinary teams and communities, and to provide person-centred personalised care. They need to be able to decipher pain narratives and have the confidence to explore psychological, social, cultural, and environmental factors alongside biomedical factors to co-produce meaningful support with empathy and cultural sensitivity.

Preparing physiotherapy students with knowledge and skills to undertake meaningful person-centred consultations and provide high-quality pain care is challenging. There are high demands on curricula, time constraints, and limited clinical placements, and this restricts opportunities for students to build the knowledge and skills needed to work with people experiencing pain, especially when the pain has been long-standing and resistant to treatment. Moreover, there is a ‘theory-practice gap’ where theoretical knowledge does not always translate into clinical practice [4,12,13]. Previously, we have argued for ‘active learning’, to shift pain education from being theory-dense to practically focused where students can contextualise and practice decision-making in a ‘safe space’. Clinical simulation is a mechanism by which this may be achieved.

Underpinned by experiential learning theory [14], clinical simulation (or simulation-based education) is an interactive educational approach that mimics an aspect of clinical practice. Simulation can be ‘high’ or ‘low’ fidelity depending on the degree to which it mirrors or replicates real clinical scenarios. Examples of simulation-based education include actors, role play, standardised or simulated patients, human computerised manikins, virtual reality, and case studies [15].

Simulation-based learning has gained increasing popularity in healthcare education as a mechanism to bridge the gap between theoretical knowledge and clinical practice [16]. Simulation is used in many areas of health education to provide learners with the opportunity to engage in lifelike scenarios that mirror clinical settings and is reported to enable the exercise of clinical reasoning, communication skills, and decision-making within a controlled environment or ‘safe space’ [17].

In the context of pain education, simulation can provide dynamic scenarios where healthcare students can practice pain assessment and management to refine and consolidate their skills. Recent literature has explored its application across various healthcare disciplines, including physiotherapy, interprofessional, nursing, and medical curricula. For example, studies have demonstrated the effectiveness of simulation in physiotherapy education for improving clinical decision-making and attitudes toward managing chronic pain [18,19]. Additionally, interprofessional simulations have been shown to enhance attitudes, beliefs, and collaboration in chronic pain management [20].

A systematic review and meta-analysis that included 51 trials (4696 participants) by the Digital Health Education Collaboration provided mixed evidence that virtual patients are at least as effective as traditional education in improving knowledge, clinical reasoning, and procedural skills with global applicability [21]. Virtual patient simulations offer educational flexibility and can be used in a variety of settings and contexts (e.g., before, during, or after classroom sessions, tutor supervised or alone, for learning and/or assessment). Moreover, virtual patients allow students to develop patient interaction skills in a ‘safe place’, which is especially useful for building confidence in learners prior to real-person consultation.

### 1.1. Aim

The aim of this project was to combine interactive simulation technology with a holistic approach to pain assessment to provide an innovative and practical tool for training healthcare students in person-centred pain care. This article reports the development, pilot, and evaluation of the virtual pain education tool with physiotherapy students.

### 1.2. Objectives

The objectives of our project were to:(a)Develop a virtual patient to simulate a clinical scenario that fosters a person-centred (biopsychosocial) line of enquiry during a consultation focusing on assessment, diagnosis and management of non-specific chronic low back pain.(b)Pilot the tool with pre-registration physiotherapy students to evaluate their ability and confidence to follow a person-centred (biopsychosocial) line of inquiry(c)Capture the attitudes and beliefs of the students on the value of the simulation as a learning tool, including (i) virtual learning environment, (ii) authenticity of the simulation, (iii) person-centred line of inquiry

## 2. Materials and Methods

This section outlines the systematic process of developing, implementing, and evaluating the virtual clinical simulation tool. The methodology is divided into two phases: the development of the virtual patient simulation, which focuses on creating an educationally robust and authentic learning tool, and the implementation and evaluation phase, which assesses its impact in fostering a person-centred biopsychosocial approach among physiotherapy students.

### 2.1. Phase I: Development of the Virtual Clinical Simulation

The development of the virtual patient simulation involved several systematic steps designed to replicate a real-life challenging scenario that a student or new graduate might encounter during routine pain assessment practice. The key stages were:Designing and educational framework—the authorship team drew on their experience in higher education to agree on clear learning objectives that aligned the virtual tool with curriculum standards and effectively supported student learning outcomes.Character design—a realistic virtual patient with lifelike characteristics, behaviors, and medical history to enhance authenticity and engagement in the learning experience.Dialogue and question design—interactive dialogue, structured question flow, and adaptive responses were created to simulate a conversation between students and the virtual patient.Feedback and scoring system development—a structured feedback mechanism was designed that provided students with constructive insights based on their interactions, helping them refine their clinical reasoning and decision-making skills.Building the virtual tool—the e-platform was developed by integrating animation, sound, and an intuitive interface to create a user-friendly learning experience that enhances engagement and knowledge retention.

### 2.2. Phase II: Implementation and Evaluation

#### 2.2.1. Implementation

The virtual patient tool was piloted with a cohort of 62 second-year physiotherapy students (43 BSc and 19 MSc) over one semester. Both cohorts of students had received academic instruction on pain management but had no clinical practice experience. Students completed the simulation during a 3-h on-campus session. During the session, students completed the virtual patient simulation and took part in a debriefing discussion led by a class facilitator.

The Lesson plan took the following format

▪ Step 1: Introduction to the session (10 min)

Briefing and opportunity for students to ask questions.

▪ Step 2: Working ‘alone’ with the virtual patient (30 min)

Students were provided with the link to the virtual patient and were asked to complete the simulation independently (wearing headphones) within a 30-minute timeframe.

▪ Step 3: Working in small groups (30 min)

Small group activity of 5–7 students. Students completed the simulation again as a group, discussing why each question is person-centred. The groups take notes.

▪ Step 4: Educator-led full class debrief (45 min)

The tutor facilitated a class discussion, with students referring to the notes they had taken in the small group activity. The facilitator encouraged students to reflect on their decisions throughout the simulation by asking open-ended questions like, “What made you choose that question?” and “How do you think the conversation might change if you approached it differently?”. This helped students think critically about their approach and the implications of their choice of questions during pain assessment.

#### 2.2.2. Evaluation

Survey Development and Pilot Testing: The survey was designed to evaluate students’ confidence in selecting person-centred biopsychosocial questions and their attitudes toward the virtual simulation’s value and impact. Likert scale items were developed to measure confidence and perceived effectiveness, while open-text questions allowed for qualitative feedback. Content validity was ensured through expert review by two educators and clinicians who were not part of the authorship/project team. Minor revisions were made based on feedback before implementation with the student cohort.

Data Collection: Students rated their confidence in selecting person-centred biopsychosocial questions using a 5-point Likert scale. A self-administered paper-based questionnaire captured student attitudes toward the virtual simulation’s value and impact using a mix of Likert scale and open-text responses. The anonymous survey method encouraged honest feedback, mitigating potential bias.

Data Analysis: The primary aim was to explore feasibility, describe preliminary outcomes, and inform future research directions rather than to establish statistical significance. However, quantitative survey data were analysed using simple descriptive statistics. Qualitative open-text responses were analysed inductively to identify key themes. Thematic analysis followed an established approach of close reading of comments to identify relevant segments, coding segments into categories, and developing a thematic model to summarise findings [22,23]. Consensus was reached by comparison and discussion of data between three authors at two separate data meetings to check that the themes accurately represented the data and research objectives.

## 3. Results

### 3.1. Phase I: The Virtual Clinical Simulation

#### 3.1.1. Educational Framework Design

The agreed learning objectives for the simulation were:Students will conduct an interactive holistic clinical consultation with a virtual person experiencing chronic pain.Students will engage in a clinical consultation that explores biopsychosocial factors contributing to their pain experience.Students will practice in a safe, controlled, and supportive environment.

The simulation was developed so that it could be undertaken in a 3-h in-person on-campus session, facilitated by two academic members of staff who delivered the brief, practicalities of supporting students to access the simulation, and the debrief.

#### 3.1.2. Character Design

Based on experiences and insights from patients and clinical staff, a fictional character was developed to reflect the motivations and concerns of service users. This provided the virtual patient with a defined background, family life, hobbies, and interests, the needs of which are balanced against health conditions as the student’s progress through the scenario. Interactive media was used to convey the patient in a realistic way. Character animation provided expression and gestures, enhancing the realism of the simulation. Audio voiceovers provide speech, emotion, and tone. These elements help students better understand and assess patient pain responses, making the learning more effective.

#### 3.1.3. Dialogue and Question Design

On starting the simulation, students were presented with a brief case history. Students were tasked to work through a series of sixteen subjective questions—at each stage, they had to choose between asking the virtual patient a question that they thought would lead to a person-centred discussion while the alternative encouraged more biomedically focused conversation (Figure 1). Students were advised that there is no ‘right or wrong’ line of questioning when assessing someone with chronic pain, but often, subtleties in question phrasing can lead to a different conversation. Students were tasked with trying to choose a line of questioning that opened up a person-centred conversation. Some of the questions were developed with a mix of straightforward options, while others required students to navigate the subtle differences in phrasing. The learning objectives were crafted to help students refine their clinical decision-making, communication skills, and ability to build rapport with patients, equipping them with practical skills they could apply in real clinical settings. The dialogue was purposefully written to create some debate and discussion in the debrief.

#### 3.1.4. Technical Considerations

The scenario was built using software that students could access on a PC, laptop, smartphone of tablet. This allowed students to experience the resource on devices they may own or be familiar with, providing greater scalability than using specialised equipment such as a VR headset. Simplifying the technical outcomes in-turn reduced the complexity of the production process, saving significant time. Building a foundation that pulled together; existing educational tools, tools free for educational use and open-source resources, provided a production workflow completed in-house at little cost. Using familiar tools, such as Articulate Storyline [24] reduced the skills-gap needed to address more complex tasks, such as character animation and speech developed with Unity [25], AutoDesk Character Generator [26] and Salsa Lipsync [27]. 

The technology was chosen balancing cost and availability of resources, whilst also considering an enjoyable interactive experience. The technology was chosen to facilitate a learning experience that made interactions intuitive and straightforward. Feedback to learners was incorporated into the technology, so that students could reflect on this both privately and as part of the session debrief. The feedback was incorporated to reinforce the learning outcomes.

#### 3.1.5. Feedback and Scoring System Design 

At the end of the simulation, the virtual patient provided students with feedback depending on the subjective line of questioning that they chose (Figure 2). The feedback was designed to be constructive and to encourage students to repeat the simulation again to consolidate learning and act on feedback. Each prompt generates an audio/visual response from the patient, which is either positive or negative in nature. For example, students selecting the non-biomechanical prompt of “Tell me a bit more about how the pain impacts your day-to-day life” generates a positive response from the virtual patient. Here, the patient responds well to the empathy shown and becomes more open, sharing the impact their pain has on family life. This helps students understand how the line of questioning can build trust between themselves and the patient. In contrast, selecting the biomechanical prompt of “I will explain a bit more about disc degeneration at the end of the assessment. First, tell me, what makes your pain worse?” will generate a negative response. The patient becomes overwhelmed and more confrontational, signaling a disconnect with the clinician’s approach.

The scenario provides one of three possible outcomes based on how well students identify non-biomechanical options. A high score of 70% and over generates a positive outcome, with the patient indicating satisfaction with the consultation and openness to advice. A low score of 30% or less results in a negative response, where the patient is frustrated and ends the consultation abruptly. Scores between 31% and 69% generate a mixed response, with the patient continuing treatment but with less enthusiasm. After the consultation, the simulation provides feedback highlighting the student’s good practices and areas for improvement, delivered by the patient through audio and video animation. Students then write a short reflection on their learning and download a document that includes their reflection, overall score, and patient feedback (Figure 3).

### 3.2. Phase II: Implementation and Evaluation

#### 3.2.1. Demographics

A total of 62 physiotherapy students attended the on-campus session to take part in the virtual patient simulation; 49 students consented to participate in the evaluation (79% survey response rate). Forty-six students provided demographic information (74% response rate): BSc *n* = 29, MSc *n* = 17; age bracket ranged from 18–21 years *n* = 23, 22–34 years *n* = 21, 35–44 years (*n* = 2); ethnicity Asian *n* = 1, Black or African American *n* = 2, White British *n* = 28, White other *n* = 2, Other *n* = 1, Prefer not to say *n* = 2; gender female *n* = 32, male *n* = 14 (Table 1).

#### 3.2.2. Frequency & Confidence in Selecting Person-Centred Questions

Analysis of survey responses demonstrated that overall, most students opted for a person-centred, biopsychosocial approach during their consultation with the virtual patient. Across the 16 items in the subjective assessment, the percentage of students selecting the ‘person-centred’ item ranged from 100% for Question 1 to 44.9% for Question 11, indicating that some questions were more challenging to interpret. This pattern was mirrored in how confident students were that they had picked the person-centred option (e.g., 89.8% of students reported high confidence in selecting the person-centred option for Question 1, whereas only 16.3% did so for Question 11).

#### 3.2.3. Analysis of Attitudes Towards the Value and Impact of Simulation

The Likert questionnaire responses demonstrated that most students found the learning experience to be valuable, indicated by the responses below, which are detailed in Table 2:(1)I enjoyed interacting with Paul [the virtual pain patient]—89.8% ‘strongly’ or ‘somewhat’ agreed(2)Interacting with Paul improved my understanding of complex subjective pain assessment—81.7% ‘strongly’ or ‘somewhat’ agreed(3)Interacting with Paul increased my confidence in dealing with complex subjective pain assessment—85.7% ‘strongly’ or ‘somewhat’ agreed(4)Interacting with Paul helped my understanding of the impact of biopsychosocial versus biomedical pain assessment—79.6% ‘strongly’ or ‘somewhat’ agreed(5)The on-screen instructions were clear—96% ‘strongly’ or ‘somewhat’ agreed(6)I could see how the activity would be relevant to my learning about pain—93.8% ‘strongly’ or ‘somewhat’ agreed(7)Paul was a believable character—93.9% ‘strongly’ or ‘somewhat’ agreed(8)The scenario seemed authentic—93.9% ‘strongly’ or ‘somewhat’ agreed(9)Practicing online with Paul gave me a safe space to practice my skills—93.9% ‘strongly’ or ‘somewhat’ agreed(10)Practicing online with Paul will help me prepare for clinical placement—87.8% ‘strongly’ or ‘somewhat’ agreed

Students had mixed views on whether they felt that they would like to repeat the simulation several times and whether the questions were challenging enough:(1)I would like to repeat the activity with Paul several times—53% ‘strongly’ or ‘somewhat’ agreed(2)I found selecting person-centred questions challenging—42.8% ‘strongly’ or ‘somewhat’ agreed

#### 3.2.4. Qualitative Analysis of Student Responses to Open Text Questions

Analysis of the attitudes and beliefs on the value of the virtual patient as a learning tool, collected by the survey questionnaire, resulted in the construction of four overarching themes [28] (1) Environmental factors & preferences, (2) Learning experience, (3) Professional development, and (4) Limitations of the simulation (Figure 4).

### 3.3. Environmental Factors & Preferences

#### 3.3.1. Focusing Attention

Students frequently emphasised the importance of environmental factors such as focus, comfort, and convenience when engaging with the virtual simulation. A quiet environment was consistently highlighted as essential for maintaining concentration and effectively engaging with the learning material. The absence of distractions was seen as crucial in allowing students to process information, reflect on their responses, and immerse themselves in the scenario:

“*Somewhere in a quiet environment, where you can focus on the questions with no distractions.*” (BSc Physiotherapy student, Male, 22–34 years)

Students also discussed the importance of a ‘private’ space. Some participants noted that being in a secluded setting (such as at home) may reduce external pressures or the fear of being observed by peers or instructors.

“*Somewhere quiet, at home, may allow people to answer more honestly. But the classroom can also be quiet too. Some people may not want others to watch what they’re doing.*” (MSc Physiotherapy student, Female, 22–34 years)

These findings align with previous research suggesting that learning environments significantly impact student engagement and performance.

#### 3.3.2. Social Interaction & Learning Dynamics

While some students preferred a private space for completing the simulation, others valued the classroom environment for its facilitation of group discussions, peer feedback, and opportunities for clarification. Engaging in the simulation within a shared learning space allowed students to reflect collectively, reconsider their choices based on peer input, and gain deeper insights through discussion:

“*In the classroom, it is really good to get feedback from other people and to change/rethink options.*” (MSc Physiotherapy student, Male, 22–34 years)

Some students found the structured classroom setting more conducive to focus and engagement. The ability to complete the simulation with instructor guidance and the chance to ask questions after the activity was particularly beneficial for enhancing understanding:

“*I liked coming in and completing [it] in the classroom with headphones as I find it easier to focus in person. However, it’s possible to complete it at home. It’s also good to have someone explain the simulation and have the opportunity to ask questions after.*” (BSc Physiotherapy student, Male, 18–21 years)

A key benefit of the classroom setting was the post-simulation debrief, where students could share experiences, clarify uncertainties, and engage in structured group reflection. This reflective process was seen as enhancing learning by allowing students to compare their own reasoning with that of their peers and gain alternative perspectives:

“*It was useful to have the class discussion afterwards.*” (BSc Physiotherapy student, Female, 18–21 years)

### 3.4. Learning Experience

#### 3.4.1. Engaging & Realistic

Students frequently described the interaction with the virtual patient as realistic, which enhanced their engagement with the simulation. The dynamic nature of the experience provided a practical, immersive way to apply clinical skills, allowing students to refine their decision-making and questioning techniques in a realistic yet low-risk environment.

“*It gave me ideas of questions to ask a complex pain patient as I feel it is hard to know what to say sometimes or how to word questions.*” (BSc Physiotherapy student, Female, 18–21 years)

Compared to traditional learning methods, such as case studies or discussions, students found the interactive nature of the simulation to be more engaging and beneficial for developing their clinical reasoning. Being able to actively participate in a consultation, rather than passively learning about it, helped them build confidence in their approach to patient interactions:

“*Interacting with a real patient was more beneficial than just talking about it.*” (MSc Physiotherapy student, Male, 22–34 years)

Students also compared the virtual simulation to traditional case-based learning, highlighting its superior ability to replicate real-life clinical encounters. Unlike static written cases, the interactive format made learning more relatable and engaging, helping students visualize and internalize key concepts more effectively:

“*It made it more relatable than a written case study. Seemed more like a real patient.*” (BSc Physiotherapy student, Female, 18–21 years)

Students reported that the simulation supported their understanding of subjective assessments in complex pain cases. By engaging with the virtual patient, they were able to explore different ways of building rapport and trust as a crucial skill in person-centred care:

“*It helped me understand a subject assessment of complex pain. Also helped me understand different ways of questioning to build trust.*” (BSc Physiotherapy student, Male, 18–21 years)

#### 3.4.2. Valuable Feedback and Debrief

Feedback emerged as a key component of the learning experience, with students valuing both the feedback provided by the virtual patient and discussions with peers and educators. The combination of automated feedback from the simulation and interactive group reflection appeared to enhance their understanding of effective questioning techniques and patient interactions.

“*The discussion with others in big groups, is very valuable, especially then getting clinical feedback from the teacher.*” (BSc Physiotherapy student, Female, 18–21 years)

Personalised feedback from the virtual patient allowed students to understand how their questioning and communication choices impacted the patient’s response. This real-time insight into how specific phrasing or inquiry styles were perceived helped students refine their approach to person-centred communication:

“*The feedback at the end from Paul, as it points out how specific lines of enquiry may have made him feel.*” (BSc Physiotherapy student, Female, 18–21 years)

Students recognised the practical relevance of the feedback to their future clinical practice, noting that the insights gained could directly inform their approach to placement:

“*The feedback at the end was very good and can help me on placement.*” (MSc Physiotherapy student, Male, 22–34 years)

#### 3.4.3. Safe Space to Build Confidence

The interactive nature of the simulation provided students with a new and alternative learning format, distinct from traditional teaching methods. Students valued the opportunity to engage in patient interactions in a ‘safe’ environment, where they could practice their questioning techniques and refine their approach without the pressures of a real-life consultation. This safe space for practicing appeared to help build confidence before encountering real patients.

“*Experience without a real person avoids real-life initial interaction being ‘messy’—gives us experience that allows us to have ideas of what to ask in our head before real-life interactions.*” (MSc Physiotherapy student, Male, 22–34 years)

Students appreciated the authenticity of the virtual patient’s responses, which made the experience feel more immersive and real. The ability to interact dynamically with a patient rather than simply reading a case study enhanced the sense of realism.

“*Paul is interactive with my answers and his story seems authentic with the details.*” (BSc Physiotherapy student, Female, 18–21 years)

A key benefit of the simulation was its ability to expose students to diverse patient reactions and emotional responses, allowing them to reflect on how their questioning style influenced the consultation process. By observing how the virtual patient reacted to different approaches, students gained a deeper understanding of rapport-building, communication strategies, and person-centred care:

“*That ‘Paul’ had more realistic emotional responses to questions. This helped to see and understand how the way questions are asked could impact the assessment rapport with the patient.*” (MSc Physiotherapy student, Female, 22–34 years)

### 3.5. Professional Development

#### 3.5.1. Development of Soft Skills

The simulation provided students with valuable experience in formulating open-ended questions that focus on a patient’s feelings, needs, and preferences rather than solely on clinical symptoms. Through interacting with the virtual patient, students recognised the importance of active listening and person-centred questioning in fostering trust and rapport.

“*Listening to his concerns is the most important thing as [it] allows for a better rapport to be built up.*” (BSc Physiotherapy student, Male, 18–21 years)

By engaging with the virtual patient, students gained a deeper appreciation of the emotional toll of chronic pain and the complexities patients face in their daily lives. This experience reinforced the importance of understanding a patient’s lived experience rather than focusing solely on clinical diagnosis or treatment plans. Students recognised that a holistic approach, which considers a patient’s perceptions, coping strategies, and personal goals, is fundamental to effective assessment and management.

“*It is important to ask about the patient, their understanding, lifestyle, how they cope, and what they want from the assessment rather than jumping straight to a diagnosis.*” (BSc Physiotherapy student, Female, 18–21 years)

#### 3.5.2. Self-Reflection & Recognition of Own Biases

Engaging with the virtual patient helped some students reflect on their own biases and clinical tendencies, fostering a deeper understanding of person-centred care. The simulation provided an opportunity for students to recognise the impact of their questioning style and approach, particularly in shaping the patient’s experience and the therapeutic relationship.

“*‘Paul’ had realistic emotional responses to questions. This helped to see and understand how the way questions are asked could impact the assessment rapport with the patient.*” (MSc Physiotherapy student, Male, 22–34 years)

The simulation highlighted the need for active listening and giving patients space to express their concerns, reinforcing the role of empathy and patient perspective in effective communication.

“*The importance of letting patients have time to speak about their experience.*” (MSc Physiotherapy student, Female, 22–34 years)

The virtual interaction prompted self-reflection on clinical biases, with some students acknowledging their tendency to focus on a biomedical model of pain rather than a broader biopsychosocial approach. This is a crucial step in developing a more holistic and person-centred perspective.

“*How much of a tendency I have towards a biological model of pain/treatment/assessment.*” (BSc Physiotherapy student, Male, 22–34 years)

### 3.6. Limitations of the Simulation

#### 3.6.1. Not Challenging Enough

While students found the simulation valuable, some expressed a desire for greater complexity and depth in the questioning options. Several students noted that the available choices sometimes felt too obvious or repetitive, limiting opportunities for critical thinking and nuanced decision-making.

“*Sometimes it was too obvious which was the ‘correct’ question.*” (BSc Physiotherapy student, Female, 22–34 years)

In some cases, students felt that multiple response options seemed equally valid, making it difficult to discern the intended learning objective. This lack of clear differentiation between options occasionally led to confusion rather than productive clinical reasoning.

“*Some of the questions felt very similar, like both could have been correct, (e.g., we have talked about rating pain 1–10), but the other answer also seemed more acceptable.*” (BSc Physiotherapy student, Female, 18–21 years)

Some students found that the progression of questions lacked coherence, with certain question pairs feeling disconnected. This made it challenging to follow a logical assessment pathway, reducing the sense of realism and flow in the consultation.

“*Some pairs of questions seemed to not link well, therefore difficult to choose which route to go down.*” (MSc Physiotherapy student, Male, 22–34 years)

A recurring suggestion was for a greater variety of question options to allow for a broader exploration of psychosocial aspects. Expanding the range of available responses could help students develop a deeper understanding of person-centred questioning by providing more diverse examples of effective communication strategies.

“*There is only 2 questions to pick from, maybe more variety would give you more examples of how to understand what a psychosocial question is.*” (MSc Physiotherapy student, Female, 22–34 years)

Students also highlighted the potential for a more dynamic, adaptive questioning structure, where responses influence subsequent questions. They suggested that incorporating a branching dialogue system where the patient’s responses shape the direction of the conversation would create a more realistic and immersive learning experience.

“*The questions could have led down different ‘paths’ instead of following questions not changing depending on the last question/answer.*” (BSc Physiotherapy student, Male, 22–34 years)

#### 3.6.2. Technical Challenges

While students generally appreciated the realism of the virtual patient, several reported technical issues that impacted their overall experience. Glitches such as freezing, crashing, and buffering were sources of frustration, occasionally disrupting the flow of interaction and requiring students to restart the activity.

“*The webpage kept freezing so I had to refresh it and start again, but not sure if this was due to my laptop or the way the activity was set up.*” (BSc Physiotherapy student, Female, 18–21 years)

In some instances, lagging performance, potentially due to high concurrent usage, was noted as a barrier to engagement. These delays may have hindered immersion and interaction, reducing the effectiveness of the simulation as a seamless learning tool.

“*Maybe try to fix technical issues with the glitching.*” (BSc Physiotherapy student, Male, 18–21 years)

Some students suggested enhancements to the user interface to improve usability. One common request was for the ability to revise an answer if selected unintentionally, allowing for more flexibility and reducing frustration when navigating the activity.

“*Opportunity to go back if accidentally clicked answer you deem incorrect.*” (BSc Physiotherapy student, Female, 18–21 years)

## 4. Discussion

The aim of this project was to combine interactive simulation technology with a holistic approach to pain assessment and provide an innovative and practical tool for training healthcare students in person-centred pain care. The findings demonstrate that students were able to distinguish person-centred from biomedically focussed questions to enable them to follow a person-centred (biopsychosocial) line of inquiry when practicing a consultation with a virtual patient experiencing chronic pain.

The findings from our study corroborate the findings of others who have evaluated the use of simulation in physiotherapy education. For instance, systematic reviews and meta-analyses of simulation research demonstrate the positive effects of simulation in physiotherapy education [29]. The use of simulations for clinical decisions during interaction with people with low back pain has been evaluated by experimental research design, though simulated patients were used over a virtual scenario [18]. However, the effects of simulation on ‘softer’ skills, such as communication, are less clear [30]. We found that students reported that they enjoyed the virtual simulation, that it increased their understanding of the complexity of people presenting with chronic pain, such as non-specific chronic low back pain, and that it increased their confidence in clinical consultations in preparation for clinical placement. The additional value of using virtual patients over standardised patients or actors is the possibility of repeated exposure to the same scenario, in an environment where it is possible to make mistakes ‘in private’. The findings from the evaluation phase of the project add to a growing body of evidence that virtual simulation can be a valuable teaching and learning resource to facilitate the learning of pre-registration healthcare students on a variety of topics [31]. It also provides valuable insight into the potential of virtual simulation to provide a ‘safe learning environment’ to prepare students with knowledge and skills to assess, diagnose, treat, and care for people with chronic pain.

Of note, students commented on the value of the class debrief and discussion with the facilitator, which helped them reflect on question choices and understand some of the ‘nuances’ in the consultation. Group discussion and debriefing played a critical role in reinforcing learning by allowing students to explore subtle differences between questions, a finding consistent with other studies highlighting the importance of debriefing in simulation-based education. This finding is consistent with others who have also demonstrated that debriefing is an essential part of simulation learning [31,32].

In the context of pain education, student feedback was consistent about the benefits of developing knowledge and skills to support a person-centred line of inquiry, especially for people with chronic pain. Importantly, the virtual simulation opened an in-class discussion about clinical situations, both within and beyond pain, where a focus on biomedical and biomechanistic lines of inquiry are common and appropriate yet still need to be conducted within a person-centred framework of care that focuses on ‘growing health’ through a social connection that involves ‘consulting the whole-person not just their condition’ [33,34,35] and for biopsychosocial pain education to underpin this as part of pre-registration training [4,36,37].

The evaluation provided valuable insights into how virtual simulations can be improved to better support student learning in person-centred pain assessment. Some students felt the scenario could be more challenging, while others saw value in starting with simpler scenarios as a foundation for complex pain assessment and online learning. The simulation utilised a forced-choice format, requiring students to distinguish between two statements. This differs from real-life clinical consultations, where healthcare professionals engage in dynamic, unscripted conversations that probe, support, and co-create understanding and care strategies with patients. As such, virtual simulations serve as an elementary introduction to consultation principles in a structured, safe learning environment. We attempted to balance the trade-off between recreating authentic real-life experiences and presenting students with an environment simple enough to facilitate learning.

Simulation-based education follows a learner-centred philosophy and a blended approach to learning that comprises elements of behaviourism, cognitivism, constructivism, and experiential learning theory [17]. It should provide a safe environment where participants can learn from their mistakes without any danger to patients, analyse and respond to realistic situations, and develop clinical knowledge, skills, behaviour, and attitudes. It also has a crucial role in improving the quality of care for patients, providing it adheres to relevant standards, such as those provided by the Association for Simulated Practice in Healthcare (ASPiH) or The International Nursing Association for Clinical Simulation and Learning INACSL [38,39].

This simulation has already been embedded as a 2–3 h in-person classroom session within the pain education curriculum of our pre-registration physiotherapy BSc and MSc courses at our higher education institution. We are in the process of developing learning materials to support post-session independent remote learning and plan to develop materials to support an entirely distance-learning version of the virtual patient.

Furthermore, we plan to use this simulation as a ‘blueprint’ to create virtual simulations for different clinical contexts and other interprofessional learning; we are working to create intermediate and more advanced virtual simulations and to develop further robust evaluation to investigate whether these virtual simulations improve knowledge and skills that are translated into clinical practice.

### 4.1. Future Scope

The generic design features of this virtual patient can be used as a blueprint for the design of elementary virtual patient simulations across interprofessional learning, contexts, settings, situations and consultation models according to need (e.g., traditional medical, biopsychosocial, person-centred, disease-illness etc.). Intermediate and advanced virtual patient simulations could utilise technological tools such as virtual reality headsets like Meta Quest (Reality Labs, a division of Meta Platforms, Menlo Park, CA, USA) or mixed reality devices such as Microsoft HoloLens (Microsoft, Redmond, WA, USA), although consideration would need to be given to increasing costs and constraining utility, such as software and delivery platform. Additionally, the integration of artificial intelligence (AI) can significantly enhance the development and utilisation of virtual patient simulations in clinical education and clinical practice. At the time of development of this project, artificial intelligence was not used as widely as it is now and, therefore, was not used in its development. The use of AI to develop future scenarios may well be a useful and efficient way to broaden and make scenarios more challenging.

### 4.2. Strengths and Limitations of the Project

A key strength of this project was the realism of the virtual patient and the clinical scenario, which was co-created by a diverse team of patients, clinicians, educators, and learners. The development process was relatively short (six months) and cost-effective, with the majority of expenses allocated to service user, student, and clinician involvement rather than software development. The e-learning platform was built in-house by a course designer with expertise in online learning integration, digital curriculum enhancement, and student digital capability development.

The evaluation used a mixed-methods approach to explore student perceptions of the simulation’s utility in introducing person-centred pain consultation. The combination of qualitative and quantitative data allowed for targeted refinements in future iterations. However, some limitations must be acknowledged. The study relied on self-reported data, which may introduce bias in how students perceived and reported their experiences. Additionally, the relatively small sample size and data collection from a single university limit the generalizability of the findings to other settings.

Nevertheless, this study provides ‘proof of concept’ that virtual simulations using a forced-choice selection method can be a valid introductory approach to teaching students how to support person-centred inquiry in clinical consultations. Importantly, the mixed-method evaluation revealed that virtual simulations also serve as a catalyst for deeper discussions about future learning needs.

## 5. Conclusions

This virtual patient simulation enhanced physiotherapy students’ ability to practice person-centred (biopsychosocial) pain assessments in a safe and supportive environment. This is a valuable tool to bridge gaps in traditional pain education by offering a realistic, interactive learning environment. Incorporating simulation into pain education to promote more empathetic and holistic patient care directly addresses key recommendations from international pain organisations such as the International Association for the Study of Pain and the European Pain Foundation.

From a clinical practice perspective, these insights support the integration of virtual simulations into pre-registration physiotherapy training to help prepare students for real-world consultations. Future directions for integrating virtual patients into pain education may include expanding the simulation’s scope to cover other critical areas of research such as sociocultural adaptations in healthcare, incorporating advanced technologies such as artificial intelligence for personalised learning experiences, and conducting longitudinal studies to assess the long-term impact on clinical decision making in practice. Longitudinal studies would also provide deeper insights into whether the skills gained through virtual simulation translate into improved patient outcomes in clinical settings.

## Figures and Tables

**Figure 1 healthcare-13-00750-f001:**
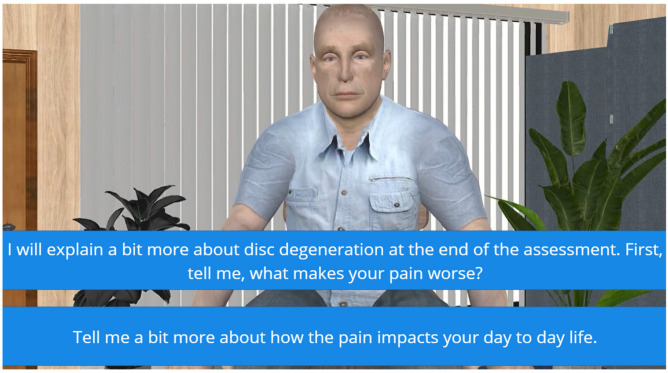
In this example, the virtual patient has told the student that he has ongoing back pain due to disc degeneration. At this point in the assessment, students must choose between focusing on disc degeneration and factors that aggravate the pain or exploring how pain impacts the virtual patient’s daily life. During the debrief, the discussion highlights that the second option is more likely to initiate a conversation about what is important to the virtual patient, thus encouraging a more person-centred approach. Students are reassured that the first option is not ‘wrong’, but it is more likely to lead to a conversation focused on biomedical factors contributing to the perception of pain. In the given scenario, the virtual patient has already received this information multiple times from other healthcare providers, and therefore, it does not open up new possibilities.

**Figure 2 healthcare-13-00750-f002:**
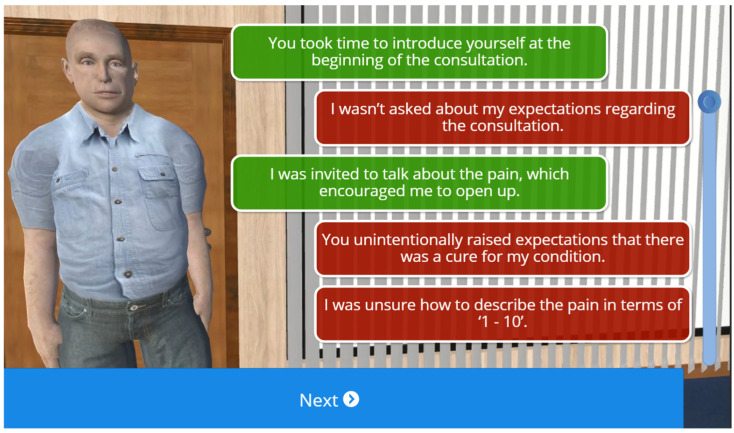
Patient feedback. At the end of the scenario, the virtual patient gives the student feedback about the impact of some of the choices of questions. The purpose of this part of the scenario is for students to appreciate the consultation from the perspective of the patient, including how some questions might be perceived differently to how they were intended. The purpose is to stimulate discussion in the debrief. Students are encouraged to challenge some of the responses and discuss alternatives.

**Figure 3 healthcare-13-00750-f003:**
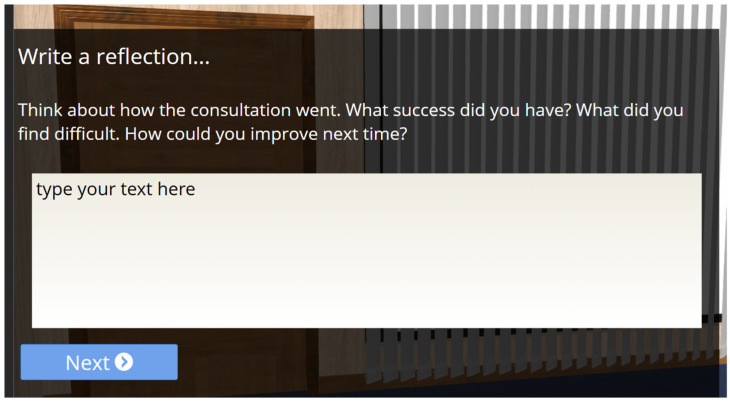
Reflection. At the end of the simulation, students are tasked with writing a reflection on their experience of interacting with the virtual patient and thinking about what they might do differently.

**Figure 4 healthcare-13-00750-f004:**
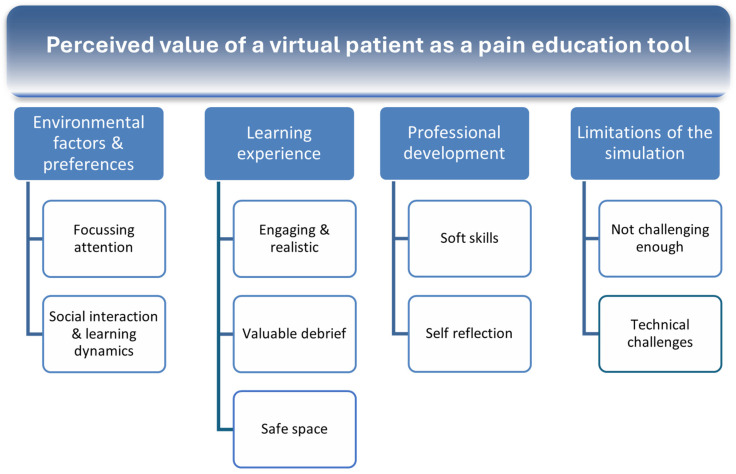
Perceived value of a virtual patient as a pain education tool.

**Table 1 healthcare-13-00750-t001:** Demographic characteristics of respondents (*n* = 49).

Demographic Category	Subcategory	Number of Respondents (*n* = 46)	Percentage (%)
Degree Program	BSc	29	59.2%
	MSc	17	34.7%
	Missing Data	0	0.0%
Age Bracket	18–21 years	23	50.0%
	22–34 years	21	45.7%
	35–44 years	2	4.3%
	Missing Data	3	6.1%
Ethnicity	Asian	1	2.0%
	Black or African American	2	4.1%
	White British	38	77.6%
	White Other	2	4.1%
	Other	1	2.0%
	Prefer not to say	2	4.1%
	Missing Data	3	6.1%
Gender	Female	32	65.3%
	Male	14	28.6%
	Missing Data	3	6.1%

**Table 2 healthcare-13-00750-t002:** Summary of survey responses to Likert scale questions.

Question	Response (*n* = 49)	Frequency	Percentage
I enjoyed interacting with Paul	Strongly agree	22	44.9
	Somewhat agree	22	44.9
	Neither agree nor disagree	3	6.1
	Somewhat disagree	0	0
	Strongly disagree	1	2
	Missing data	1	2
I found selecting person-centred questions challenging	Strongly agree	3	6.1
	Somewhat agree	18	36.7
	Neither agree nor disagree	7	14.3
	Somewhat disagree	17	34.7
	Strongly disagree	2	4.1
	Missing data	2	4.1
Interacting with ‘Paul’ improved my understanding of complex subjective pain assessment	Strongly agree	16	32.7
	Somewhat agree	24	49
	Neither agree nor disagree	6	12.2
	Somewhat disagree	1	2
	Strongly disagree	0	0
	Missing data	2	4.1
Interacting with ‘Paul’ increased my confidence in dealing with complex subjective pain assessment	Strongly agree	5	10.2
	Somewhat agree	37	75.5
	Neither agree nor disagree	3	6.1
	Somewhat disagree	2	4.1
	Strongly disagree	1	2
	Missing data	1	2
Interacting with ‘Paul’ helped my understanding of the impact of biopsychosocial v biomedical pain assessment	Strongly agree	18	36.7
	Somewhat agree	21	42.9
	Neither agree nor disagree	7	14.3
	Somewhat disagree	1	2
	Strongly disagree	1	2
	Missing data	1	2
The on-screen instructions were clear	Strongly agree	38	77.6
	Somewhat agree	9	18.4
	Neither agree nor disagree	0	0
	Somewhat disagree	0	0
	Strongly disagree	1	2
	Missing data	1	2
I could see how the activity would be relevant to my learning about pain	Strongly agree	28	57.1
	Somewhat agree	18	36.7
	Neither agree nor disagree	2	4.1
	Somewhat disagree	0	0
	Strongly disagree	0	0
	Missing data	1	2
Paul was a believable character	Strongly agree	29	59.2
	Somewhat agree	17	34.7
	Neither agree nor disagree	1	2
	Somewhat disagree	1	2
	Strongly disagree	0	0
	Missing data	1	2
The scenario seemed authentic	Strongly agree	29	59.2
	Somewhat agree	17	34.7
	Neither agree nor disagree	1	2
	Somewhat disagree	1	2
	Strongly disagree	0	0
	Missing data	1	2
Practicing online with Paul gave me a safe space to test my skills	Strongly agree	37	75.5
	Somewhat agree	9	18.4
	Neither agree nor disagree	1	2
	Somewhat disagree	0	0
	Strongly disagree	0	0
	Missing data	2	4.1
Practicing online with Paul will help me prepare for clinical placement	Strongly agree	16	32.7
	Somewhat agree	27	55.1
	Neither agree nor disagree	3	6.1
	Somewhat disagree	2	4.1
	Strongly disagree	0	0
	Missing data	1	2
I would like to repeat the activity with Paul several times	Strongly agree	11	22.4
	Somewhat agree	15	30.6
	Neither agree nor disagree	11	22.4
	Somewhat disagree	6	12.2
	Strongly disagree	3	6.1
	Missing data	3	6.1
The online feedback helped me understand what areas I needed to improve on	Strongly agree	17	34.7
	Somewhat agree	12	24.5
	Neither agree nor disagree	3	6.1
	Somewhat disagree	2	4.1
	Strongly disagree	1	2
	Missing data	14	28.6

## Data Availability

The data supporting the findings of this study are available from the corresponding author upon reasonable request.

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
