# Peer review of "Development, Implementation, and Evaluation of a ‘Virtual Patient’ with Chronic Low Back Pain: An Education Resource for Physiotherapy Studentsâ€"

_healthcare, 2025, doi:10.3390/healthcare13070750_

Round 1
Reviewer 1 Report
Comments and Suggestions for Authors
Thank you for the opportunity to review this manuscript. This work addresses a highly relevant and important topic in healthcare education. The study investigates an area of growing importance in education, especially given the need to better prepare healthcare professionals for holistic and empathetic patient care.
The manuscript is mostly clear, but several sections require reorganisation for coherence. Some aspects of the method and results sections need to be better delineated, as there is overlap that detracts from the clarity of the findings. Strengthening the discussion and addressing some inconsistencies in data presentation would further enhance the manuscript’s impact.
Abstract: Authors report that 62 students participated in the simulation, but only 49 completed the evaluation. Including the response rate would provide a clearer understanding of participant engagement and data completeness.
Introduction: The introduction is well-written and provides a useful overview of the field. However, the statement on lines 99-100, “Literature in this field has predominantly focussed on interprofessional (17), nursing (18) and medical curricula,” cites older references (2011). More recent literature, especially within the last decade, should be explored to ensure the introduction reflects current trends and developments.
Methods: Objectives (Lines 118-127): These are not methods and should be moved earlier in the manuscript as aims and objectives.
Authors should outline the phases of development, implementation, and evaluation more explicitly to guide readers through the methodology.
Phase I: Development of the VP - More detail is required on the VP. The authors state that this is high fidelity - what makes this so? Also, can the authors provide an example of the kind of responses that were provided by the VP? Was this only a selection of one/two statements? was this all in text or text-to-audio? How were responses provided back to students? Consider including examples of the types of responses provided by the VP to illustrate the interaction, and adding images or screenshots of the VP interface to enhance understanding.
Phase II: Lines 170-171: Clarify the differences in educational backgrounds between MSc and BSc students. How might their prior training influence their interaction with the VP?
Lines 180-181: The lesson plan, referred to as Supplementary File 2, seems to be incorrectly labelled. This could be incorporated into the text for clarity. Details on how the debrief was facilitated would be useful —e.g., what techniques were used, and how was reflection encouraged?
Data Collection (Lines 183-186): Supplementary File 2 is cited as the 16-item questionnaire but this is attached as the evaluation questionnaire. Clarify whether these represent two distinct tools or aspects of the same questionnaire, as it’s unclear how students rated confidence on a 5-point Likert scale.
Specify how the survey was distributed (online or paper) and whether it was validated or tested for reliability.
Data analysis: The rationale for entering data into SPSS when no statistical analyses were performed is unclear.
Ethical approval: The subtitle appears misplaced at the end of the previous section (Line 203).
Results: The results section is somewhat disorganised and should follow the phases outlined in the methods for consistency.
Phase I: Avoid including implementation/evaluation data (e.g., Table 1) here as it pertains to Phase II.
Phase II: Clearly separate demographic data, evaluation results, and confidence ratings.
Response rate needs to be explicitly stated.
Section 3.1.1 and 3.1.2 contain details that belong in the methods section.
Lines 252-253: The calculation of participants (19 BSc + 17 MSc = 36, not 46) needs correction.
Figure 1: The description is insufficient. Explain the differences between blue and orange bars and consider removing redundant elements for clarity.
Section 3.2.3: Questions repeated in Supplementary File 2 are unnecessary here.
Line 296: it’s unclear what citation 21 is citing here – these are the researchers results
Qualitative Results: Include participant details (e.g., MSc/BSc or other defining characteristics) to contextualise findings.
Discussion: The discussion requires more critical analysis. The first paragraphs mostly reiterate results without deeper interpretation.
Lines 472-474: These ideas could be incorporated into a “Future Scope” section, highlighting the potential for embedding the VP in curricula.
Strengths and limitations: No limitations are currently discussed. Address potential issues such as sample size, generalisability, and the use of self-reported data.
Lines 511 and 515-517: Ensure references are consistent with the chosen citation style.
Conclusion: The conclusion should be more impactful. Summarise the key findings, their significance for healthcare education, and the potential future directions for VP integration.
Lines 532-537: do not belong in the conclusion.
Author Response
Thank you for the detailed and constructive review of our manuscript. Please find below a point by point reply to the comments. We have made major amendments throughout the paper and uploaded an amended 'clean' copy of the manuscript. A tracked copy is available if needed to see where amendments have been made.
Thank you for the opportunity to review this manuscript. This work addresses a highly relevant and important topic in healthcare education. The study investigates an area of growing importance in education, especially given the need to better prepare healthcare professionals for holistic and empathetic patient care.
The manuscript is mostly clear, but several sections require reorganisation for coherence. Some aspects of the method and results sections need to be better delineated, as there is overlap that detracts from the clarity of the findings. Strengthening the discussion and addressing some inconsistencies in data presentation would further enhance the manuscript’s impact.
Abstract: Authors report that 62 students participated in the simulation, but only 49 completed the evaluation. Including the response rate would provide a clearer understanding of participant engagement and data completeness.
Authors’ Response:
Thank you for the detailed and constructive review of our manuscript. We agree with all suggestions and have made major amendments throughout.
The methods and results sections have been amended to more clearly demarcate information.
Response rate: 49/62x100 = 79% has been added.
Introduction: The introduction is well-written and provides a useful overview of the field. However, the statement on lines 99-100, “Literature in this field has predominantly focussed on interprofessional (17), nursing (18) and medical curricula,” cites older references (2011). More recent literature, especially within the last decade, should be explored to ensure the introduction reflects current trends and developments.
Authors’ Response:
This section has been amended to include more up to date citations.
Methods: Objectives (Lines 118-127): These are not methods and should be moved earlier in the manuscript as aims and objectives.
Authors’ Response:
These have been moved
Authors should outline the phases of development, implementation, and evaluation more explicitly to guide readers through the methodology.
Authors’ Response:
This section has been amended throughout to focus on methodology.
Phase I: Development of the VP - More detail is required on the VP. The authors state that this is high fidelity - what makes this so? Also, can the authors provide an example of the kind of responses that were provided by the VP? Was this only a selection of one/two statements? was this all in text or text-to-audio? How were responses provided back to students? Consider including examples of the types of responses provided by the VP to illustrate the interaction, and adding images or screenshots of the VP interface to enhance understanding.
Authors’ Response:
We have added more information to explain how the virtual patient was developed
Phase II: Lines 170-171: Clarify the differences in educational backgrounds between MSc and BSc students. How might their prior training influence their interaction with the VP?
Authors’ Response:
An extra line has been added to clarify that both cohorts of students had received academic instruction on pain management but had no clinical practice experience. The difference in educational backgrounds was not collected as part of this study. It is a requirement of the MSc course that students have a previous degree, though some students on the BSc course also have these prior experiences.
Lines 180-181: The lesson plan, referred to as Supplementary File 2, seems to be incorrectly labelled. This could be incorporated into the text for clarity. Details on how the debrief was facilitated would be useful —e.g., what techniques were used, and how was reflection encouraged?
Authors’ Response:
The supp file has been removed and the content added to the main text under ‘methods – implementation’. Additional information has been added to this same section about the debrief.
Data Collection (Lines 183-186): Supplementary File 2 is cited as the 16-item questionnaire but this is attached as the evaluation questionnaire. Clarify whether these represent two distinct tools or aspects of the same questionnaire, as it’s unclear how students rated confidence on a 5-point Likert scale.
Authors’ Response:
Students were asked to rate their confidence that they had chosen the more biopsychosocial question for each of the 16 items. Supp file 2 has been changed.
Specify how the survey was distributed (online or paper) and whether it was validated or tested for reliability.
Authors’ Response:
This information has been added.
Data analysis: The rationale for entering data into SPSS when no statistical analyses were performed is unclear.
Authors’ Response:
Thank you for highlighting this. The manuscript has been amended to avoid confusion.
Ethical approval: The subtitle appears misplaced at the end of the previous section (Line 203).
Authors’ Response:
This section has been moved to the end of the manuscript.
Results: The results section is somewhat disorganised and should follow the phases outlined in the methods for consistency.
Authors’ Response:
The results section has been edited and reorganised throughout to follow the phases of the methods section.
Phase I: Avoid including implementation/evaluation data (e.g., Table 1) here as it pertains to Phase II.
Authors’ Response:
This section has been moved whilst reorganising the results section.
Phase II: Clearly separate demographic data, evaluation results, and confidence ratings.
Authors’ Response:
These sections have been reorganised throughout whilst rewriting the methods and results sections
Response rate needs to be explicitly stated.
Authors’ Response:
The response rate has been added.
Section 3.1.1 and 3.1.2 contain details that belong in the methods section.
Authors’ Response:
This section has been reorganised throughout.
Lines 252-253: The calculation of participants (19 BSc + 17 MSc = 36, not 46) needs correction.
Authors’ Response:
This typo has been corrected – thank you
Figure 1: The description is insufficient. Explain the differences between blue and orange bars and consider removing redundant elements for clarity.
Authors’ Response:
Figure 1 has been replaced.
Section 3.2.3: Questions repeated in Supplementary File 2 are unnecessary here.
Authors’ Response:
Supplementary file 2 has been replaced
Line 296: it’s unclear what citation 21 is citing here – these are the researchers results
Authors’ Response:
This has been amended along with the reorganisation of results
Qualitative Results: Include participant details (e.g., MSc/BSc or other defining characteristics) to contextualise findings.
Authors’ Response:
This information has been added to the qualitative comments.
Discussion: The discussion requires more critical analysis. The first paragraphs mostly reiterate results without deeper interpretation.
Authors’ Response:
The discussion has been rewritten throughout.
Lines 472-474: These ideas could be incorporated into a “Future Scope” section, highlighting the potential for embedding the VP in curricula.
Authors’ Response:
This section has been addressed whilst rewriting the discussion. A new subheading ‘future scope’ has been added.
Strengths and limitations: No limitations are currently discussed. Address potential issues such as sample size, generalisability, and the use of self-reported data.
Authors’ Response:
This section has been rewritten to provide more balance on strengths and limitations.
Lines 511 and 515-517: Ensure references are consistent with the chosen citation style.
Authors’ Response:
These have been updated
Conclusion: The conclusion should be more impactful. Summarise the key findings, their significance for healthcare education, and the potential future directions for VP integration.
Lines 532-537: do not belong in the conclusion.
Authors’ Response:
The conclusion has been expanded with the above points addressed
Reviewer 2 Report
Comments and Suggestions for Authors
I am grateful for the opportunity to review the manuscript entitled "Development, Implementation and Evaluation of a ‘Virtual Patient’ with Chronic Low Back Pain: an education resource for healthcare students." This paper addresses a highly relevant area, highlighting issues that significantly impact patient outcomes.
Nevertheless, the report can be improved in some aspects. We stress that the merit of your work is not in question; however, some aspects need clarification or reformulation.
Title: the study was developed exclusively with physiotherapy students. The title should reflect that. The use of "healthcare students" may be inaccurate.
Abstract: please comply with the author's guidelines, the word count is over 250. Results should detail quantitative results according to the study's objective. Identify how this study addresses limitations in existing educational tools.
Keywords: With the same rationale as the title, the word multidisciplinary may not be appropriate.
Introduction: There are parts of the text that miss the appropriate acknowledgment (e.g. ln. 48-51; ln. 68-74; ln. 81-84; ln. 105-111). The introduction should be redirected to physiotherapy students and evidence related to pain management by physiotherapists, considering that the scope of interventions on pain management is distinct across healthcare professions. The study was developed exclusively with physiotherapy students. Authors should focus on the specific contribution of this work for physiotherapists and patients. The framework should emphasize these aspects. It is not clear which specific gap in the evidence the authors intended to address. Integrating recent meta-analyses or systematic reviews to provide updated evidence for the efficacy of virtual simulation in education may be useful. Present the limitations of current pain education to focus more on virtual patient simulations.
Methods: Please justify the choice of mixed-methods design and explain its appropriateness for evaluating educational interventions.
Results: results are hard to read. A more straightforward approach, aligned with the objectives of the study, should be considered.
Discussion: The first paragraph repeats the results. Provide a shorter overview of the results to drive the discussion. The authors could relate findings to experiential learning theory or other educational models to strengthen theoretical underpinnings. Include a more robust discussion of how findings compare with existing literature to better contextualize the study’s contributions.
Conclusions: What does this paper add to current evidence? And implications for clinical practice? Please provide concrete recommendations for scaling or further research.
Author Response
Thank you for the detailed and constructive review of our manuscript. Please find below a point by point reply to the comments. We have made major amendments throughout the manuscript and uploaded a revised 'clean' copy. A tracked copy is available if helpful to see where amendments have been made.
I am grateful for the opportunity to review the manuscript entitled "Development, Implementation and Evaluation of a ‘Virtual Patient’ with Chronic Low Back Pain: an education resource for healthcare students." This paper addresses a highly relevant area, highlighting issues that significantly impact patient outcomes.
Nevertheless, the report can be improved in some aspects. We stress that the merit of your work is not in question; however, some aspects need clarification or reformulation.
Title: the study was developed exclusively with physiotherapy students. The title should reflect that. The use of "healthcare students" may be inaccurate.
Authors’ Response:
Thank you for the detailed and constructive review of our manuscript. We agree with all suggestions and have made major amendments throughout.
Abstract: please comply with the author's guidelines, the word count is over 250. Results should detail quantitative results according to the study's objective. Identify how this study addresses limitations in existing educational tools.
Authors’ Response:
The abstract has been revised
Keywords: With the same rationale as the title, the word multidisciplinary may not be appropriate.
Authors’ Response:
The context has been addressed within the revised abstract
Introduction: There are parts of the text that miss the appropriate acknowledgment (e.g. ln. 48-51; ln. 68-74; ln. 81-84; ln. 105-111). The introduction should be redirected to physiotherapy students and evidence related to pain management by physiotherapists, considering that the scope of interventions on pain management is distinct across healthcare professions. The study was developed exclusively with physiotherapy students. Authors should focus on the specific contribution of this work for physiotherapists and patients. The framework should emphasize these aspects. It is not clear which specific gap in the evidence the authors intended to address. Integrating recent meta-analyses or systematic reviews to provide updated evidence for the efficacy of virtual simulation in education may be useful. Present the limitations of current pain education to focus more on virtual patient simulations.
Authors’ Response:
The introduction has been revised throughout, to direct to physiotherapy training.
Methods: Please justify the choice of mixed-methods design and explain its appropriateness for evaluating educational interventions.
Authors’ Response:
This has been added to the study strengths/limitations section.
Results: results are hard to read. A more straightforward approach, aligned with the objectives of the study, should be considered.
Authors’ Response:
The results have been rewritten throughout and better aligned with methods/objectives.
Discussion: The first paragraph repeats the results. Provide a shorter overview of the results to drive the discussion. The authors could relate findings to experiential learning theory or other educational models to strengthen theoretical underpinnings. Include a more robust discussion of how findings compare with existing literature to better contextualize the study’s contributions.
Authors’ Response:
The discussion has been expanded and rewritten to address both reviewers comments.
Conclusions: What does this paper add to current evidence? And implications for clinical practice? Please provide concrete recommendations for scaling or further research.
Authors’ Response:
The conclusion has been expanded to address these points.
Round 2
Reviewer 1 Report
Comments and Suggestions for Authors
Thank you for your revisions. I have identified several areas for further refinement. Below are specific points for consideration:
Abstract:
Some minor language typos:
- Self-reported
- Desire for more complexity
- Ensure that the themes presented in the abstract align with those in the results section (see L325-328).
Introduction:
L91 - enable "s" - remove the s
Methods:
- Lesson Plan - consider writing "intro" in full.
- The methods section does not provide much information about the questionnaire, while more details appear in the results section. Consider shifting this information to the methods.
- What is meant by the term "fluidity" ? Also, clarify whether the design involved a multiple-choice selection between two options.
L232 had chosen
L283 - not clear what students attended
L316-322: Text says 50% and 40% and then later 36.2% - Please ensure numerical consistency and specify the correct percentage.
L325-328: Themes don't match what was previously given in abstract
Explain in methods whether consensus was reached for the themes and how this was determined. Expand on the subthemes to clarify their distinctions.
Results
- Consider removing quotes that essentially convey the same point but from different participants, particularly in the first theme and subtheme.
- Themes 1b and 2b appear to express similar ideas—consider whether they should be merged or more clearly differentiated.
- L463 The term "high fidelity" is not typically applied to virtual patients in the same way as mannequins. Please clarify what aspect of the VP makes it high fidelity.
-L527: ASPiH and ANACSL are used but not previously defined
Author Response
Authors response: Reviewer 1
Thank you for your revisions. I have identified several areas for further refinement. Below are specific points for consideration:
Authors’ Response:
Thank you for taking the time to review and provide constructive feedback on our manuscript. Your feedback has significantly enhanced the clarity and presentation of our study.
Abstract:
Some minor language typos:
- Self-reported
- Desire for more complexity
- Ensure that the themes presented in the abstract align with those in the results section (see L325-328).
Authors’ Response:
Many thanks for identifying these – the suggested refinements have been addressed in the manuscript.
Introduction:
L91 - enable "s" - remove the s
Authors’ Response:
Thank you
Methods:
- Lesson Plan - consider writing "intro" in full.
- The methods section does not provide much information about the questionnaire, while more details appear in the results section. Consider shifting this information to the methods.
- What is meant by the term "fluidity" ? Also, clarify whether the design involved a multiple-choice selection between two options.
L232 had chosen
L283 - not clear what students attended
L316-322: Text says 50% and 40% and then later 36.2% - Please ensure numerical consistency and specify the correct percentage.
L325-328: Themes don't match what was previously given in abstract
Explain in methods whether consensus was reached for the themes and how this was determined. Expand on the subthemes to clarify their distinctions.
Authors’ Response:
Thank you for your recommendations. I have addressed each point in the manuscript and uploaded a tracked-changes version for your review. The subthemes have been expanded and tied to the comment in the results section below.
Results:
- Consider removing quotes that essentially convey the same point but from different participants, particularly in the first theme and subtheme.
- Themes 1b and 2b appear to express similar ideas—consider whether they should be merged or more clearly differentiated.
- L463 The term "high fidelity" is not typically applied to virtual patients in the same way as mannequins. Please clarify what aspect of the VP makes it high fidelity.
-L527: ASPiH and ANACSL are used but not previously defined
Authors’ Response:
Thank you for your recommendations. I have addressed each point in the manuscript and uploaded a tracked-changes version for your review. The description of the subthemes have been refined/updated and tied to the comment in the methods section above.
Reviewer 2 Report
Comments and Suggestions for Authors
I would like to thank the authors for their willingness to accept and incorporate the suggestions made.
Author Response
Authors’ Response: Reviewer 2
I would like to thank the authors for their willingness to accept and incorporate the suggestions made.
Authors’ Response:
Thank you for taking the time to review and provide constructive feedback on our manuscript. Your feedback has significantly enhanced the clarity and presentation of our study.